# Generation of internal solitary waves by frontally forced intrusions in geophysical flows

Daniel Bourgault[1,2], Peter S. Galbraith[3] & Cédric Chavanne[1]

Internal solitary waves are hump-shaped, large-amplitude waves that are physically analogous to surface waves except that they propagate within the fluid, along density steps that typically characterize the layered vertical structure of lakes, oceans and the atmosphere. As do surface waves, internal solitary waves may overturn and break, and the process is thought to provide a globally significant source of turbulent mixing and energy dissipation. Although commonly observed in geophysical fluids, the origins of internal solitary waves remain unclear. Here we report a rarely observed natural case of the birth of internal solitary waves from a frontally forced interfacial gravity current intruding into a two-layer and vertically sheared background environment. The results of the analysis carried out suggest that fronts may represent additional and unexpected sources of internal solitary waves in regions of lakes, oceans and atmospheres that are dynamically similar to the situation examined here in the Saguenay Fjord, Canada.

[1] Institut de sciences de la mer de Rimouski, Université du Québec à Rimouski, 310 allée des Ursulines, Rimouski, Québec, Canada G5L 3A1. [2] Laboratoire de mécanique des fluides et d'acoustique (École Centrale de Lyon, CNRS, Université Lyon 1, INSA Lyon), 69134 Lyon, France. [3] Institut Maurice Lamontagne, Fisheries and Oceans Canada, 850 route de la Mer, Mont-Joli, Canada G5H 3Z4. Correspondence and requests for materials should be addressed to D.B. (email: daniel_bourgault@uqar.ca).

nternal solitary waves[1–3] are hump-shaped nonlinear and nonhydrostatic gravity waves that propagate horizontally along density waveguides in stratified geophysical fluids. Although patchy and episodic, these waves are known to be widespread and energetic enough to participate in global-scale air and water mass modifications, as they may break into stratified turbulence when encountering destabilizing conditions[4]. Although these waves, once formed, are routinely observed in lakes[5], oceans[3,6] and the atmosphere[7], their origin often remains elusive. The current understanding is that these waves can be generated by stratified flows over topography[8–14], by subharmonic interactions[15], by internal tide, seiche and Kelvin wave steepening[5,16,17], by tidal beams impinging on the pycnocline[18,19] or, and importantly for this paper, by the gravitational collapse of mixed fluids, equivalently referred to as intrusive gravity currents[20–24].

While the gravitational collapse of mixed fluids mechanism has received considerable research attention from the theoretical and experimental points of view since the 1980s, there has been little clear field evidence to support whether this mechanism occurs in geophysical fluids in general, and in oceans in particular. One exception for the ocean is the reporting of internal solitary waves generated by the Columbia River plume flowing into the stratified coastal Pacific[25]; an upside-down analogous situation to that studied in lock-exchange laboratory experiments where dense water is suddenly left to flow underneath a two-layer still ambient[20,22]. In the atmosphere, the Morning Glory phenomenon observed over North-Western Australia is also thought to arise from the generation of internal solitary waves by an intruding bottom gravity current in the stratified ambient[26,27]. Bottom density currents produced by thunderstorm outflows have also been reported as a potential source of atmospheric internal solitary waves[28].

The generation of internal solitary waves by the gravitational collapse of mixed fluids is not limited to cases of surface (for example, river plumes) or bottom (for example, Morning Glory) density currents but may also arise, under some circumstances, from interfacial gravity currents or intrusions, in two-layer fluids, as reproduced in several lock-exchange laboratory experiments[20–24,29,30]. One important result of laboratory experiments is that large-amplitude internal solitary waves that are able to detach and lead the interfacial gravity current head are only generated when the initial motionless condition is said to be in non-equilibrium, that is, when the centre of mass of the lock water rises or falls when the gate is removed[21–23,31]. For full-depth lock-exchange experiments, that is, experiments where the lock water initially occupies the full tank depth[31], where the two layers of the ambient fluid are of equal thicknesses, equilibrium situations exist when the density of the intrusion equals the mean density of the ambient, a condition easily achieved by a complete mixing of the two-layer ambient water on the lock side of the gate. To obtain non-equilibrium initial conditions, either salt is added to the mixture or a certain volume of the mixed lock water is replaced by an equivalent volume of freshwater[22].

The generation of internal solitary waves by non-equilibrium interfacial gravity currents is not yet fully understood partly because the wide parameter space that defines this problem has only been partially explored[31]. For example, the case where the lock water initially occupies only a fraction of the total water depth, a situation referred to as partial-depth intrusion, remains little studied, although likely of greater environmental significance. In any case, whether or not non-equilibrium interfacial gravity currents are an important source of internal solitary waves in geophysical fluids remains to be demonstrated, especially since it has been suggested that non-equilibrium

geophysical situations are rare and that equilibrium situations are 'notably of greater geophysical relevance'[31].

Here we report on geophysical observations of the generation of internal solitary waves by what can be interpreted as being non-equilibrium stratification, partial-depth[31] and frontally forced interfacial gravity currents intruding into a quasi two-layer and vertically sheared ambient. This is a complex natural situation that, to our knowledge, has not been fundamentally studied theoretically or experimentally nor clearly observed in geophysical fluids before. For example, laboratory experiments on intrusive gravity currents have only considered still ambient, and non-equilibrium situations have been studied for full-depth lock-exchange problems, but hardly for partial-depth[31]. We conclude that fronts may be important sources of internal solitary waves in geophysical fluids that are dynamically similar to the situation examined here in the Saguenay Fjord, Canada (Fig. 1).

## Results

**Observations.** The essence of our echo-sounder, current and density measurements is summarized by two repeated transect lines carried out in a region of the Saguenay Fjord[32] (Fig. 2) known to host internal solitary waves of unknown origin[33]. These data were acquired $\sim 1$ h before the time of low water (20:30 UTC on 10 July 2015 at Tadoussac) during a neap ebb tide of range $\Delta\eta = 2.9$ m. The first transect (Fig. 2a–c) shows a prominent solitary wave of amplitude $a = 7.4$ m and maximum vertical velocities $w_s = \pm 0.2$ m s$^{-1}$ propagating northward along a pycnocline of buoyancy frequency $N_{\mathrm{pyc}} = 0.25$ s$^{-1}$ located at depth $h = 7.6$ m in the undisturbed state ahead. The total depth $H$ along this transect varies between 100 and 150 m (not shown). Idealized numerical simulations (presented below) indicate that such a solitary wave in this stratified and sheared background environment has horizontal lengthscale $\lambda = 50$ m and induces a perturbation mechanical energy $E' = 1.9$ MJ m$^{-1}$ (see Methods for details). This wave is therefore highly nonlinear[3] ($a/h = 1$), long relative to the surface layer thickness ($\lambda/h = 7$) but fairly short relative to the bottom layer thickness ($H - h$) that varies over the transect in the range 92–142 m such that $0.4 < \lambda/(H - h) < 0.5$. The fact that the wave is short relative to the bottom layer thickness will become important below for interpreting these observations. This wave was approximately the fourth wave seen in this area, and the previous ones seen had greater amplitude, up to $a = 10$ m (Fig. 3).

The wave of Fig. 2a is followed by what can be interpreted from the echogram as being an intermediate turbulent layer, centred $\sim 10$ m depth and being roughly 20 m thick, populated with Kelvin–Helmholtz billows near the top edge. Eleven density overturns were detected (see Methods for details) throughout this transect (Fig. 2a), of which seven were within the billowing layer behind the solitary wave. These seven overturns are characterized by mean Thorpe scales $\bar{L}_T = 0.92$ m (s.d. = 0.36 m). Assuming a 1:1 ratio between Thorpe and Ozmidov scales[34] gives an order of magnitude for the mean dissipation rate $\mathcal{O}(\bar{\epsilon}) \sim 10^{-5}$ W kg$^{-1}$ within the overturns, consistent with intense ocean turbulent conditions[17] and supporting the echogram interpretation that this layer is turbulently mixing.

These observations may suggest that the turbulent layer was produced by the passage of the wave in a manner comparable to that seen in idealized numerical simulations of unstable solitary waves where turbulent fluid is left behind as the breaking wave keeps moving forward and restabilizes. This can happen for example when the underlying bottom topography shallows sufficiently to cause wave breaking[35,36]. However, this hypothesis is eliminated here because the wave remains short along this transect relative to

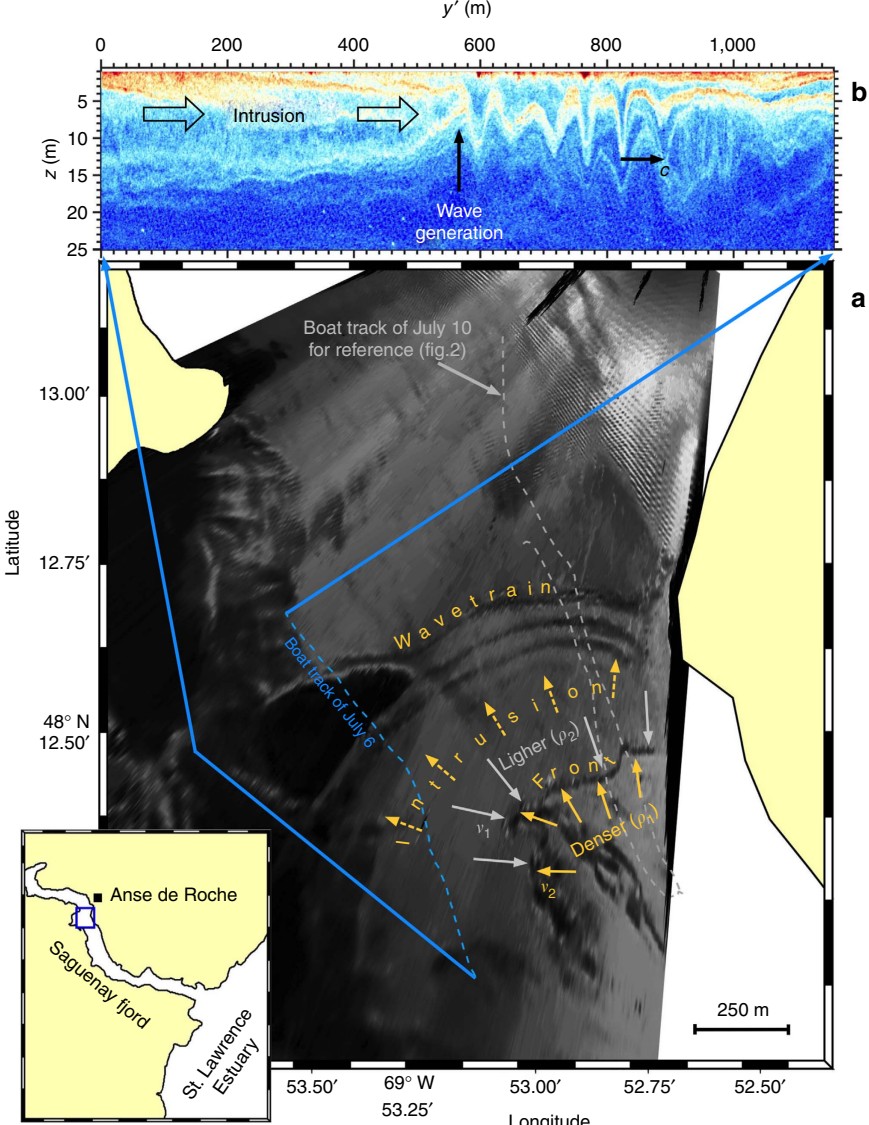

**Figure 1 | Map of the study area in the Saguenay Fjord.** (**a**) Sea surface patterns at 15:59 UTC on 6 July 2015 and (**b**) the echogram collected shortly after along the blue dashed line, between 16:11 and 16:20 UTC. The vectors (grey and yellow) and annotations represent a qualitative interpretation of the image depicting the generation of internal solitary waves by a frontally forced intrusion. The grey dashed line is the boat track of 10 July between 19:15 and 19:48 UTC (Fig. 2). The inset shows the location of Anse-de-Roche and the blue box represents the study area shown in **a**.

the bottom layer thickness ($0.4 < \lambda/(H-h) < 0.5$) such that it is virtually unaffected by the underlying changing bottom topography. This is supported by a numerical simulation we have carried out (not shown) in which the propagation of a comparable wave over the changing bottom topography was simulated. As expected, the result showed negligible change in wave shape and behaviour relative to a flat-bottom simulation.

Another possibility to explain the overturning and billowing fluid seen behind the wave could be that it was produced by shear instability that had previously developed within the solitary wave[37]. Qualitatively, parallels could well be made between the observations of Fig. 2a and the numerical results presented in Lamb and Farmer[37] (their Figs 13,14 and 18). However, there are reasons to also reject this hypothesis. First, the vorticity seen in the observed billowing layer is opposite to that expected from shear instability that would arise from such a northward propagating internal solitary wave (Fig. 4). For example, the billow centred around $y' = -275$ m and at 6 m depth rotates

anti-clockwise, while billows arising from wave-induced shear instability would be expected to rotate clockwise[37]. Another reason to reject this hypothesis is that the reduced stratification within the wave is $N^2 - S^2/4 > 0$ (equivalent to $N^2/S^2 > 1/4$)[38] such that there is no indication that this wave may have been dynamically unstable.

An alternative and more plausible interpretation is that the Kelvin–Helmholtz billows arose from the vertical shear associated with an intrusive layer that lies just above the pycnocline and characterized by northward velocity of up to $v = 0.5 \, \text{m} \, \text{s}^{-1}$ (Fig. 2b,e). This interpretation is supported by the fact that the reduced stratification averaged over the length of the intrusion $\overline{(N^2 - S^2/4)}$ is negative near the top edge of the sheared interface (Fig. 5). The Reynolds number $\text{Re} = 5 \times 10^6$ is around three orders of magnitude higher than typical laboratory settings of intrusive gravity currents in two-layer fluids[22]. This near-surface intrusion differs from deeper intrusions found in other fjords[39–42] and in the Saguenay[32] by the fact that it intrudes through the

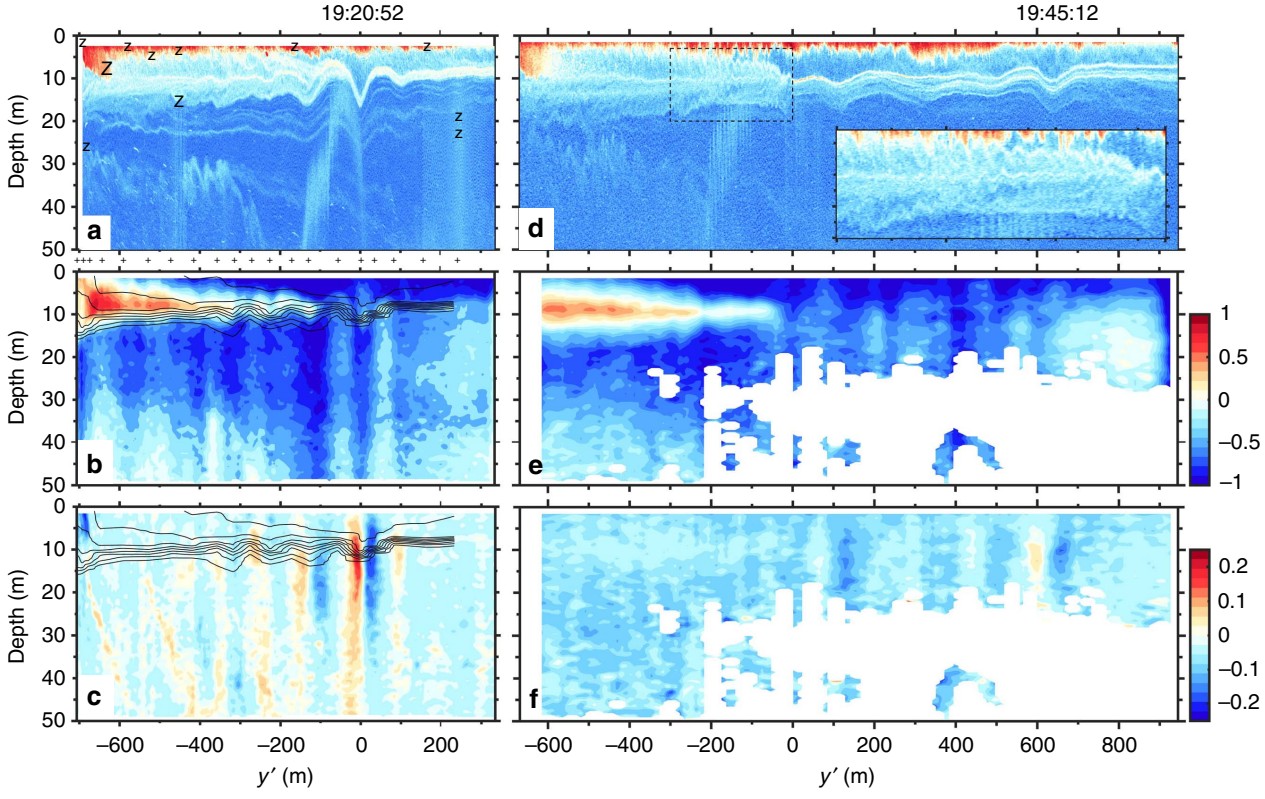

**Figure 2 | Field observations.** (**a,d**) Echograms, (**b,e**) northward velocity $v$ (ms$^{-1}$) and (**c,f**) upward velocity $w$ (ms$^{-1}$). The black contours on **b** and **c** are isopycnals 2 kg m$^{-3}$ apart based on sorted density profiles. The 'Z' symbols on **a** mark the centre of detected density overturns and their size is proportional to the logarithm of the available potential energy of the fluctuation $P$ found in the range $4 \times 10^{-6} - 3 \times 10^{-4}$ W kg$^{-1}$. The '+' symbols between **a** and **b** mark the positions of the 20 temperature–salinity profiles collected. The inset highlights Kelvin–Helmholtz billows within the dashed box. These transects correspond to the grey track presented in Fig. 1.

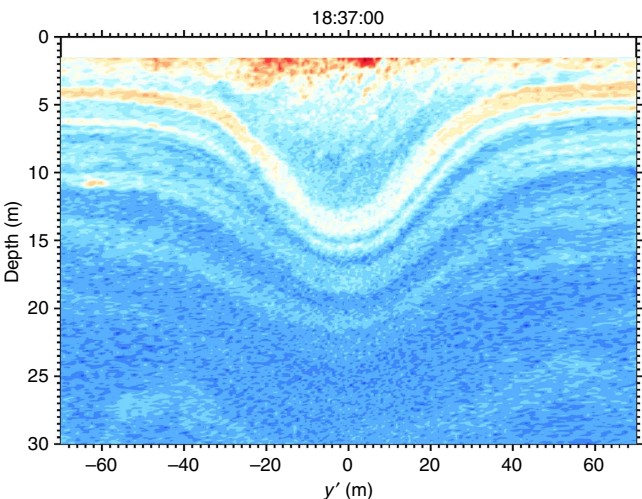

**Figure 3 | Echogram of a large internal solitary wave.** Echogram of the largest internal solitary wave ($a = 10$ m) seen during this sampling. This northward propagating wave was observed ahead of the intrusion, roughly at the same position (centred around $y' = 0$) as the wave on Fig. 2a seen $\sim 45$ min later.

sharp pycnocline where internal solitary waves are most susceptible to being excited.

The observations suggest that this dynamically unstable intrusion was forced by a small-scale convergent front found at the southernmost end of the transect ($y' = -680$ m; Figs 2 and 4). This front is characterized by steep isopycnals angled at 25°, strong backscatter intensity indicative of bubble entrainment typical of such fronts[43], a large downward vertical velocity $w_{front} = -0.2$ m s$^{-1}$ within the first 10 m or so of the water column and of width $L \approx 20$ m, a surface density jump $\Delta \rho = 4$ kg m$^{-3}$ and Rossby number $R_O = v/(fL) = 2 \times 10^2 \gg 1$, where $f = 1.08 \times 10^{-4}$ s$^{-1}$ is the Coriolis parameter and $v \approx 0.5$ m s$^{-1}$. This front falls into the type of submesoscale fronts that are ubiquitous in the coastal ocean[44,45].

By the second transect (Fig. 2d–f), the leading solitary wave previously seen at 19:20:52 UTC had detached from the head of the intrusion and was captured again 635 m northward at 19:45:12 with an amplitude $a = 4.4$ m. This wave had travelled against the surface current at an average ground speed $c = 0.4$ m s$^{-1}$. The reason for its reduced amplitude may be due to radial spreading loss as there is no evidence that this wave had lost amplitude due to wave breaking. Evidence of internal solitary wave radial spreading can be seen in this environment from shore-based cameras (for example, Fig. 1). A few other internal solitary waves of smaller amplitudes were also seen trailing behind, as if these waves had somehow been ejected from the head of the intrusion (Fig. 2d), an hypothesis that will be supported below with numerical simulations. During this second transect, the intrusion was still clearly defined and more Kelvin–Helmholtz billows were visible near the top of its sheared interface (Fig. 2d, inset).

Shore-based photography suggests that the phenomenon was phase-locked with the semi-diurnal tide as surface patterns indicative of the concomitant and proximate existence of fronts

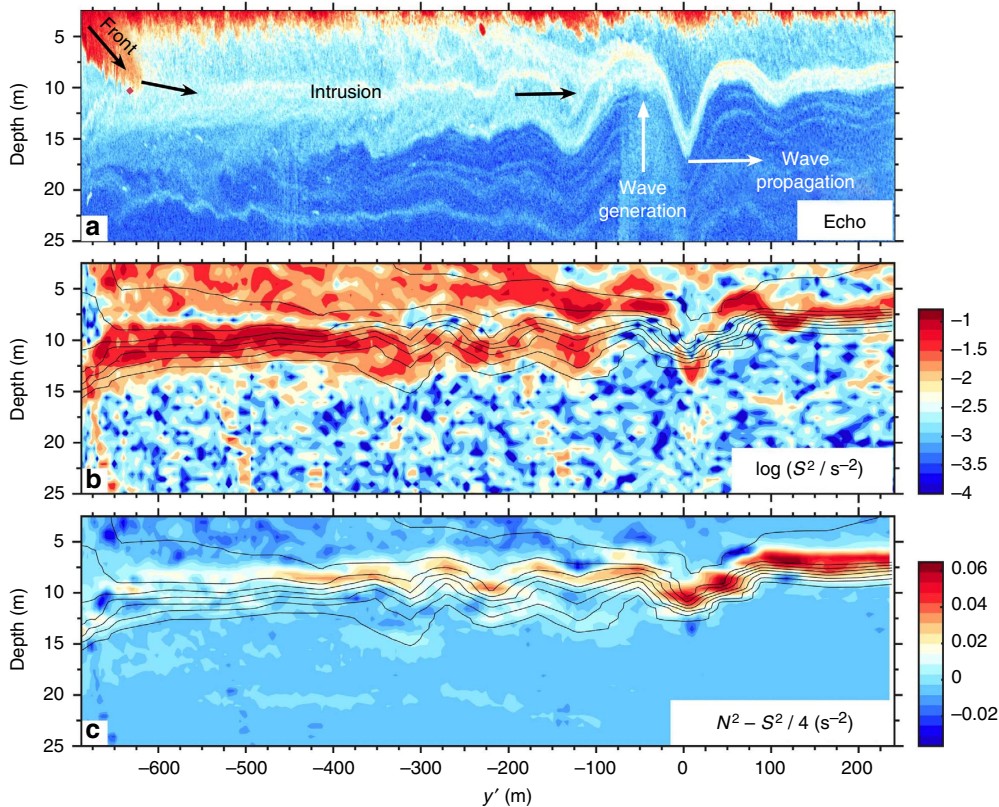

**Figure 4 | Shear and reduced stratification.** Details of the birth of an internal solitary wave from the head of the interfacial gravity current. (**a**) Echogram; (**b**) the logarithm of the shear squared $S^2$ (colour) with contoured isopycnals (black) 2 kg m$^{-3}$ apart; and (**c**) the reduced stratification $N^2 - S^2/4$ (colour) with contoured isopycnals (black).

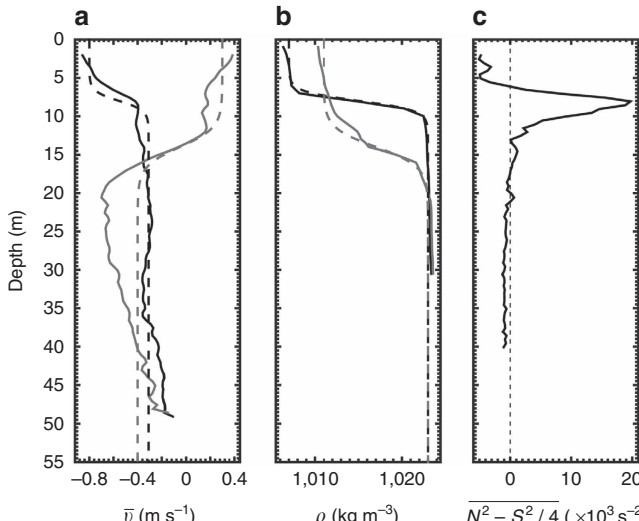

**Figure 5 | Background conditions.** Observed (solid) and idealized (dashed) background currents (**a**) and density (**b**) profiles on the north (black) and south (grey) side of the front. (**c**) Reduced stratification profile spatially averaged within the intrusion in the [−600, −100 m] interval (Fig. 2).

and internal solitary waves were seen on the 6, 7, 9 and 10 of July in the same area and around the same tidal phase, that is, an hour or so before the time of low water at Tadoussac. July 8 was too windy to capture any interpretable surface patterns. The clearest

camera and underwater evidence of this process in action was captured on the 6 (Fig. 1). Note that at the time these data were acquired, we were not aware that this process was taking place. It is by chance that we happened to carry out an echo-sounder transect at that place and time, and it is only during the subsequent data analysis phase of this research that we realized we had sampled something unusual. This also explains why the echo-sounder transect does not extend across the front on that day. At the time of sampling, we were not preoccupied by fronts. Fronts and internal solitary waves will however be the focus of future field experiments.

**Numerical simulations.** Next, we carried out numerical simulations to determine whether this unusual series of concomitant phenomena (Kelvin–Helmholtz billows, internal solitary waves, intrusion and front) seen on Fig. 2 was simply coincidental or whether they were tied together by a single process, possibly related to intrusive gravity currents. Given the number of different nonlinear and nonhydrostatic processes involved, we based our analysis on the two-dimensional incompressible Euler equations. These are the simplest equations that can explicitly deal with all salient features of this problem (see Methods for details).

A first numerical simulation was carried out in a 150 m deep flat-bottom open channel initialized with a three-layer idealization of the observed front (Figs 5 and 6a,d,g; Methods for details). Importantly for the rest of this demonstration, the ambient is not initially still but is characterized by a convergent flow at the front, and the background environment is sheared, a complex natural situation not examined before in laboratory experiments. This configuration is equivalent to a partial-depth intrusion because

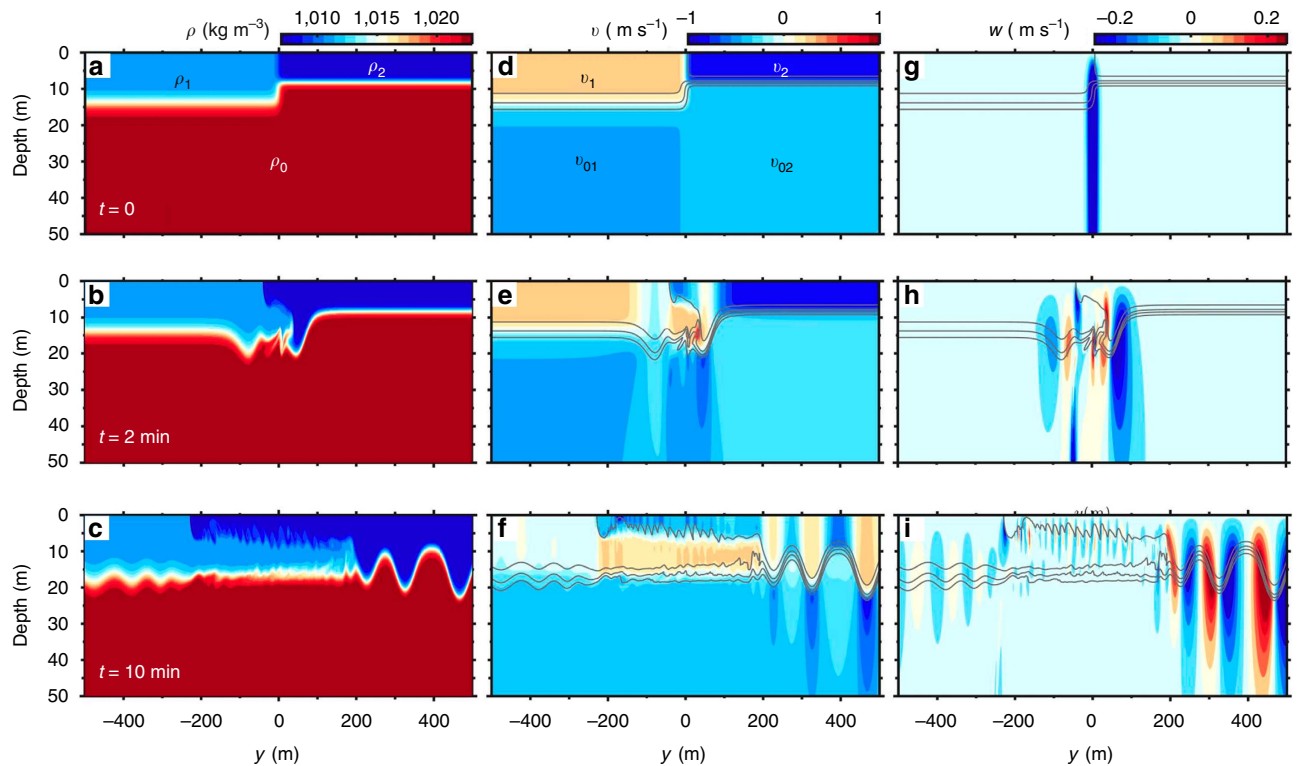

**Figure 6 | Numerical results.** Numerical simulations of Kelvin–Hemlholtz billows and internal solitary waves generated by a frontally forced intrusion. (**a–c**) Density $\rho$, (**d–f**) northward current $v$ and (**g–i**) upward current $w$ with density contours superimposed as grey lines. The top row (**a,d,g**) represents the model initialization ($t = 0$) with parameters $\rho_0$, $\rho_1$, $\rho_2$, $v_1$, $v_2$, $v_{01}$ and $v_{02}$ as defined in the Methods section and given in Table 1. Only the top 50 m are shown but the model domain is 150 m deep.

**Table 1 | Parameter values (in meter-kilogram-second) used for each simulations carried out.**

| # | $\rho_0$ | $\rho_1$ | $\rho_2$ | $v_{01}$ | $v_{02}$ | $v_1$ | $v_2$ | $h_1$ | $h_2$ | $d_1$ | $d_2$ | $L$ | $Ri_1$ | $Ri_2$ | $Fr_1$ | $Fr_2$ | $\xi$ | $\Delta$ |
|---|---|---|---|---|---|---|---|---|---|---|---|---|---|---|---|---|---|---|
| 1 | 1023 | 1011 | 1007 | −0.4 | −0.3088 | 0.3 | −0.8 | 14 | 8 | 2.50 | 1.25 | 20 | 1.2 | 1.6 | 0.26 | 0.73 | −0.18 | −0.89 |
| 2 | 1023 | 1011 | 1007 | 0 | 0 | 0 | 0 | 14 | 8 | 2.50 | 1.25 | 20 | ∞ | ∞ | 0 | 0 | −0.18 | −0.89 |
| 3 | 1023 | 1013.8571 | 1007 | 0 | 0 | 0 | 0 | 14 | 8 | 2.50 | 1.25 | 20 | ∞ | ∞ | 0 | 0 | 0 | −0.89 |
| 4 | 1023 | 1013.8571 | 1007 | −0.4 | −0.3186 | 0.2 | −0.8 | 14 | 8 | 2.50 | 1.25 | 20 | 1.2 | 1.7 | 0.21 | 0.73 | 0 | −0.89 |

See Methods for details.

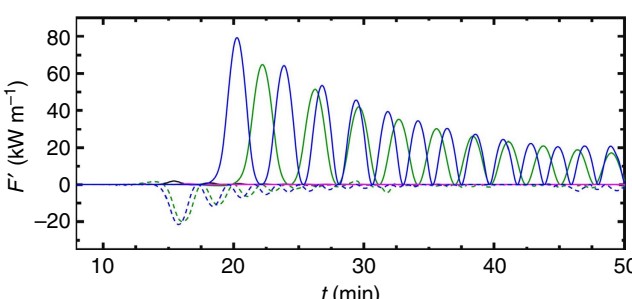

**Figure 7 | Energy fluxes.** Simulated southward (negative values) and northward (positive values) depth-integrated perturbation energy fluxes $F'$ at (dashed) $y = -1$ km and (solid) $y = 1$ km for different parameter values. The blue curves represent the control run based on an idealization of the field observations (see Fig. 6 for a more explicit representation of this run). Other curves represent sensitivity tests carried out using the parameters listed in Table 1. Blue, control run #1; black, run #2; magenta, run #3; green, run #4.

the intrusive fluid of density $\rho_1$ does not initially occupy the entire water depth[31]. For partial-depth and initially motionless intrusions, the degree of non-equilibrium can be characterized by the following non-dimensional density ratio[31]

$$\xi = \frac{g'_{12}h_2 - g'_{01}(h_1 - h_2)}{g'_{02}h_1}, \tag{1}$$

where

$$g'_{ab} = g\left(\frac{\rho_a - \rho_b}{\rho_0}\right). \tag{2}$$

Equilibrium situations correspond to $\xi = 0$. For this simulation, $\xi = -0.18$ (Table 1, run #1). The situation can also be characterized by the ratio of layer thicknesses of the ambient[22], that is, $\Delta = [h_2 - (H - h_2)]/H = -0.89$. Both interfaces on each side of the front are initially dynamically stable with minimum interfacial shear Richardson number $Ri_1 = 1.2$ on the south side (that is, $y < 0$) and $Ri_2 = 1.6$ on the north side ($y > 0$). Finally, both regions are also initially subcritical with corresponding composite Froude numbers $Fr_1 = 0.26$ and $Fr_2 = 0.73$ less than unity (Table 1, run #1).

The result of this control run shows remarkable similarities with the field observations (Fig. 6). Such a convergent front forces a dynamically unstable intrusion comparable in thickness and velocity to the observations and characterized by roughly 5 m tall Kelvin–Helmholtz billows developing on the upper interface. Internal solitary waves are generated in both directions. A few southward propagating ones with amplitudes $a < 4$ m are generated near the front during the first few minutes. The leading wave induces an energy perturbation $E' = 2.0$ MJ m$^{-1}$. These southward propagating waves may be similar to those generated by river plumes[25] with the difference that, here, wave generation does not require the flow to be initially supercritical. We cannot confirm whether such smaller amplitude southward propagating waves were also emitted in the Saguenay Fjord as sampling did not extend far enough southward of the front.

The most remarkable solitary waves are generated at the head of the intrusion and propagate northward. The waves are rank-ordered with the leading wave having, at $t = 1{,}100$ s and $y = 901$ m, amplitude $a_1 = 14.1$ m, horizontal lengthscale $\lambda_1 = 70$ m, phase speed $c_1 = 0.91$ m s$^{-1}$ and energy perturbation $E' = 7.8$ MJ m$^{-1}$. The head of the intrusion continuously emits waves northward with one wave being emitted roughly every six buoyancy periods ($\tau = 2\pi/N_2 = 25$ s). This provides a continuous northward depth-integrated perturbation energy flux $F'$ that reaches an instantaneous value of max($F'$) = 79 kW m$^{-1}$ for the leading wave and then decreases with time (Fig. 7). The total wavetrain perturbation energy induced during about 30 min ($E' = \int_{18\,min}^{50\,min} F'\,\mathrm{d}t$), which is a lower bound estimate of the events duration observed in the field, is $E' = 37.4$ MJ m$^{-1}$.

Three other simulations were carried out to explore the sensitivity of the resulting internal solitary waves to whether the initial condition is in equilibrium or not and/or the stillness of the ambient (Table 1). The first of these tests examines the situation where the initial density field is identical as the control run discussed above (that is, $\xi = -0.18$) but where the ambient is still (run #2 in Table 1). This corresponds to a non-equilibrium situation where, according to laboratory experiments, large-amplitude solitary waves leading the intrusion head are expected to be generated. This is indeed what the numerical results show but the northward propagating waves generated are at least an order of magnitude less energetic than the control run (Fig. 7, black). The second of these tests examines the still ambient with the initial density field in equilibrium (that is, $\xi = 0$, run #3 in Table 1). Virtually, no waves are generated, in agreement with laboratory experiments (Fig. 7, magenta). Finally, we examine the case where the initial density field is in equilibrium but the ambient fluid is convergent and sheared (that is, $\xi = 0$, run #4 in Table 1). Interestingly, this equilibrium situation generates very large-amplitude internal solitary waves (Fig. 7, green), much larger than the non-equilibrium and still simulation (Fig. 7, black), and almost as large and as energetic as the non-equilibrium control run (Fig. 7, blue). These results show that intruding gravity currents, when forced, can generate large-amplitude internal solitary waves even when the initial state at rest is in equilibrium.

## Discussion
The model results support the hypothesis that the observed concomitant phenomena reported here (front, intrusion, Kelvin–Helmholtz billows, internal solitary waves; Fig. 2) are not simply randomly coincidental but arise from a single physical process that could be called a forced gravitational collapse of mixed fluids or a frontally forced intrusion. We conclude from this research that partial-depth, convergent fronts in stratified,

sheared but initially dynamically stable (Ri > 1/4) and subcritical (Fr < 1) geophysical turbulent fluids (Re > 10$^6$) can force dynamically unstable intrusions that trigger Kelvin–Helmholtz billows, leading to strong turbulent dissipation and mixing, while continuously radiating very large-amplitude ($a/h \sim 1$) internal solitary waves through the head of the intrusion. While non-equilibrium stratifications ($\xi \neq 0$) favour the generation of larger amplitude waves, it is not a condition for generating large-amplitude internal solitary waves in frontally forced intrusions.

As this is a rather new problem, it still remains to be determined how the intrinsic properties of the intrusion, Kelvin–Helmholtz billows, internal solitary waves and turbulence vary across the complex and rich parameter space that defines this problem (for example, Re, Ri, Fr, Ro, $\xi$ and $\Delta$). Such relationships are required before turbulence and internal wave fluxes arising from dynamically similar geophysical intrusions and fronts as reported here could be parameterized and incorporated into large-scale models. Furthermore, historical observations should be re-examined in the light of this finding as in some circumstances fronts and forced intrusions could explain elusive field observations of internal solitary waves.

## Methods
**Observations.** A field experiment took place during 5–11 July 2015 near Anse-de-Roche in the Saguenay Fjord (Fig. 1). Measurements were collected from an 8 m research boat equipped with a towed 120 kHz Biosonics echo-sounder, a towed 500 kHz Teledyne RD Instruments 5-beam acoustic Doppler current profiler (ADCP) and a Sea-Bird Electronics SBE-19plus conductivity–temperature–depth profiler. The ADCP data were acquired in 5 s ensemble averages and 0.5 m vertical bin size. The data were then averaged to 1.0 m vertical resolution and 15 s temporal resolution. The ADCP had no bottom-tracking feature. The flow measurements were referenced to ground using global positioning system (GPS) data acquired every second. A 20-megapixel Canon EOS 6D camera with a 60 degree field of view was programmed to take one image every minute during daytime from the balcony of Mrs Simard at 48° 13.439′ N, 69° 52.513′ W and altitude of 89 m, pointing towards the southwest. Images were then georectified using landscapes and the boat as ground control points[33].

The *in situ* data presented here were collected on 6 July between 16:11 and 16:20 UTC (Fig. 1) and on 10 July 2015 between 19:15 and 19:48 UTC (Fig. 2). The first transect presented (Fig. 2a–c) was carried out while our boat was freely drifting southward with surface currents, while 20 temperature–salinity casts were collected as quickly as possible down to a depth of ~40 m to obtain the highest temporal resolution across the pycnocline. This sampling was nevertheless insufficient to fully resolve high-frequency internal solitary waves given that at a sampling fall speed of ~1 m s$^{-1}$ it took roughly the same time to carry out one profile as the buoyancy period (≈30 s). Once this sampling finished, we steamed back along more or less the same transect line while towing our echo-sounder and acoustic current profiler without performing en-route temperature–salinity casts (Fig. 2d–f), explaining the absence of density measurements on the second transect. En-route current measurements were subject to greater uncertainties due to movements of the towed body and bubble entrainment while steaming. This explains the sparsity of the current data on the second transect. This sampling sequence was guided by visual observations of sea surface patterns. Note that this sequence did not visually appear to be any different than the previous tens of internal solitary waves we had sampled during this field experiment. We had no clear real-time indications that we were sampling something unusual such that we called it a day at the end of the second transect.

The buoyancy frequency $N^2 = -(g/\rho_0)\partial\rho/\partial z$, where $\rho$ is water density, $g = 9.81$ m s$^{-2}$ is Earth's gravitational acceleration, $\rho_0 = 1{,}023$ kg m$^{-3}$ is a reference density and $z$ is the upward axis, was calculated from gravitationally sorted density profiles and decimated at 1 m vertical resolution. The vertical shear was calculated as $S^2 = (\partial u/\partial z)^2 + (\partial v/\partial z)^2$, where $u$ and $v$ are the eastward and northward velocity components, respectively.

Overturns were detected within density profiles[34], where a minimum run length of 4 was found to distinguish unstable features from noise, which were then screened for temperature and salinity co-variance. Only overturns larger than order 1 m could therefore be detected. A total of 11 were found, for which their associated dissipation rates were estimated as the available potential energy of the fluctuations multiplied by $N$ calculated on the sorted density profiles. Given that turbulence quantities are log-normally distributed, of the mean dissipation rate of turbulence kinetic energy was estimated as $\bar{\epsilon} = exp(m + s^2/2)$, where $m$ and $s^2$ are the arithmetic mean and variance of $\ln(\epsilon/(\mathrm{W\,kg}^{-1}))$, respectively[46].

**Model.** The numerical model used here solves with second-order finite differences the following two-dimensional, non-rotating Euler equations in the vertical $y$–$z$

plane with a free surface[47]:

$$\frac{\partial v}{\partial t} + \frac{\partial v^2}{\partial y} + \frac{\partial (wv)}{\partial z} = -\frac{1}{\rho_0}\frac{\partial p}{\partial y}, \tag{3}$$

$$\frac{\partial w}{\partial t} + \frac{\partial (vw)}{\partial y} + \frac{\partial w^2}{\partial z} = -\frac{1}{\rho_0}\frac{\partial p}{\partial z} - \frac{\rho}{\rho_0}g, \tag{4}$$

$$\frac{\partial v}{\partial y} + \frac{\partial w}{\partial z} = 0, \tag{5}$$

$$\frac{\partial \eta}{\partial t} = -\frac{\partial}{\partial y}\int_{-H}^{\eta} v\, dz, \tag{6}$$

$$\frac{\partial \rho}{\partial t} + \frac{\partial (v\rho)}{\partial y} + \frac{\partial (w\rho)}{\partial z} = 0, \tag{7}$$

where $t$ is time, $y$ the meridional axis (positive northward), $z$ the vertical axis (positive upward), $v$ and $w$ the horizontal and vertical velocity, respectively, $\eta$ the position of the free surface, $\rho$ the density, $\rho_0 = 1,023\,kg\,m^{-3}$ a constant reference density, and $p$ the pressure. The Poisson equation needed for solving the nonhydrostatic pressure is solved using the Pardiso solver[48–50]. Implicit numerical viscosity and diffusion associated with the second-order flux-limiter advection scheme[47] insures numerical stability.

Simulations were carried out for a flat-bottom rectangular domain of depth $H = 150\,m$, which roughly corresponds to the depth at the sampling transect. The initial density $\rho$ and horizontal velocity $v$ fields that define the convergent front were set as the following three-layer system with

$$\rho(y,z,t=0) = \rho_- + \frac{\rho_+ - \rho_-}{2}\left[1 + \tanh\left(\frac{y}{L/2}\right)\right], \tag{8}$$

where

$$\rho_- = \rho_0 + \frac{\rho_1 - \rho_0}{2}\left[1 + \tanh\left(\frac{z + h_1}{d_1}\right)\right], \tag{9}$$

$$\rho_+ = \rho_0 + \frac{\rho_2 - \rho_0}{2}\left[1 + \tanh\left(\frac{z + h_2}{d_2}\right)\right], \tag{10}$$

with $(\rho_0, \rho_1, \rho_2) = (1,023, 1,011, 1,007)\,kg\,m^{-3}$ are the density of the layers, $(h_1, h_2) = (14, 8)\,m$ are the depth of the surface layers on, respectively, the north and south side of the front, $(d_1, d_2) = (2.50, 1.25)\,m$ are the half-thickness of the pycnocline, and $L = 20\,m$ is the front width. Similarly,

$$v(y,z,t=0) = v_- + \frac{v_+ - v_-}{2}\left[1 + \tanh\left(\frac{y}{L/2}\right)\right], \tag{11}$$

where

$$v_- = v_{01} + \frac{v_1 - v_{01}}{2}\left[1 + \tanh\left(\frac{z + h_1}{d_1}\right)\right], \tag{12}$$

$$v_+ = v_{02} + \frac{v_2 - v_{02}}{2}\left[1 + \tanh\left(\frac{z + h_2}{d_2}\right)\right], \tag{13}$$

with $(v_{01}, v_{02}, v_1, v_2) = (-0.4000, -0.3088, 0.3000, -0.8000)\,m\,s^{-1}$. This initial condition is such that $\int_{-H}^{0} v_-\,dz = \int_{-H}^{0} v_+\,dz$. The initial vertical velocity $w$ is numerically computed by integrating over $z$ the continuity equation 5 given $v(y,z,t=0)$, that is

$$w(y,z,t=0) = -\int_{-H}^{z}\frac{\partial v}{\partial y}\,dz \tag{14}$$

This is an idealization of the observed conditions found on each side of the front (Figs 5 and 6a,d,g). The square of the buoyancy frequencies and shears at the pycnocline on each side of the front are, respectively, $N_1^2 = -(g/\rho_0)(\rho_1 - \rho_0)/(2d_1) = 0.023\,s^{-2}$, $N_2^2 = -(g/\rho_0)(\rho_2 - \rho_0)/(2d_2) = 0.061\,s^{-2}$, $S_1^2 = [(v_1 - v_{01})/(2d_1)]^2 = 0.020\,s^{-2}$ and $S_2^2 = [(v_2 - v_{02})/(2d_2)]^2 = 0.039\,s^{-2}$. The minimum Richardson numbers at the sheared interfaces on each side of the front are $Ri_1 = N_1^2/S_1^2 = 1.2$ and $Ri_2 = N_2^2/S_2^2 = 1.6$. Both interfaces are therefore initially dynamically stable given that $Ri > 1/4$ (ref. 38). The composite Froude numbers[51] on each side of the front are, respectively, $Fr_1 = \sqrt{v_1^2/(g_1' h_1) + v_{01}^2/[g_1'(H - h_1)]} = 0.26$ and $Fr_2 = \sqrt{v_2^2/(g_2' h_2) + v_{02}^2/[g_2'(H - h_2)]} = 0.73$, where $g_i' = 2d_i N_i^2 (i = 1, 2)$ is the reduced gravity on each side of the front, respectively. Given that these Froude numbers are $< 1$ indicates that long linear internal waves could escape and propagate in both directions on each side of the front.

These parameters listed above are for the control run. Three other simulations were carried out to test the sensitivity of the results to some change in the parameters. The parameters used for these additional runs are listed in Table 1.

The horizontal numerical domain extends in the interval $y = \pm 200\,km$. The grid size varies with highest resolution $\Delta y = 0.5\,m$ in the $[-1\,km, 1\,km]$ interval and the resolution steadily decreases to $\Delta y = 1\,km$ outside the central domain of interest. This long coarse resolution domain acts as a sponge layer that numerically absorbs barotropic waves that escape during the initial adjustment and ensures that

the solution is not affected by open boundary conditions. The grid size also varies in the vertical, with highest resolution $\Delta z = 0.5\,m$ in the $[0\,m, 30\,m]$ interval that steadily decreases to $\Delta z = 5\,m$ below. The number of grid points is $4,663 \times 103$ grid points. The time step $\Delta t$ is adjusted at every iteration to be a third of the Courant–Friedrichs–Lewy condition. Two other simulations were carried out to test the sensitivity of the results of the control run (run #1 in Table 1) to either increasing or decreasing the spatial resolution in both directions by a factor of 2, and therefore correspondingly increasing/decreasing the time step by roughly a factor of 2. Qualitatively, the low-resolution simulation appears slightly more diffuse, but the main features such as the generation of large-amplitude internal solitary waves and Kelvin–Helmholtz instability are equally well reproduced. Quantitatively, the largest amplitude leading wave produced by the low-resolution run differs by 4% from the control run. The main conclusions drawn from these simulations are therefore considered to be insensitive to grid resolution.

The depth-integrated perturbation energy density flux radiating away from the front in both directions at $y = -1\,km$ ('$-$' subscript) and $y = 1\,km$ ('$+$' subscript) is calculated as[52,53]

$$F_- = \int_{-H}^{\eta}\left[vp' + v\frac{\rho_0}{2}[v^2 + w^2] + vP\right]dz - \int_{-H}^{\eta} v_-^3\,dz, \tag{15}$$

and

$$F_+ = \int_{-H}^{\eta}\left[vp' + v\frac{\rho_0}{2}[v^2 + w^2] + vP\right]dz - \int_{-H}^{\eta} v_+^3\,dz, \tag{16}$$

where $p'$ is the sum of the perturbation hydrostatic pressure and nonhydrostatic pressure and $P$ is the available potential energy relative to the reference density profile taken either to the far left (for $F_-$) or far right (for $F_+$) of the domain[54]. These energy fluxes contain not only high-frequency fluctuations due to internal waves but also lower-frequency changes due to the varying background conditions caused by the intrusion. These trends are removed with an envelope filter before integrating the flux to obtain the perturbation energy $E' = \int F'\,dt$ where $F'$ is the high-pass filtered flux (Fig. 7).

The horizontal lengthscale of an internal solitary wave is computed as[55]

$$\lambda = \frac{1}{a}\int_{y_1}^{y_2}|\eta(y)|\,dy, \tag{17}$$

where $a$ is the wave maximum vertical displacement, $\eta$ is the position of the interface, and $y_1$ and $y_2$ are chosen to include the entire wave.

**Code availability.** The code that supports the findings of this study, written in FORTRAN 90, is available from the corresponding author on reasonable request.

**Data availability.** The data that support the findings of this study are available from the corresponding author on reasonable request.

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

## Acknowledgements

This research was funded by the National Sciences and Engineering Research Council of Canada, the Canada Foundation for Innovation, the Department of Fisheries and Oceans Canada and is a contribution to the scientific program of Québec-Océan. We thank Teledyne RD Instruments for loaning us their new V-ADCP Monitor used in this study. We also thank Alexandre Livernoche and Pascal Bourgault for their assistance in the field and data analysis of the camera data. Finally, we would also like to thank Louis Gostiaux (École Centrale de Lyon), Kraig Winters (Scripps Institution of Oceanography) and Kevin Lamb (University of Waterloo) for general discussions about internal wave dynamics and for their comments on the manuscript as well as the anonymous reviewers for their constructive critiques.

## Author contributions

D.B. co-organized (with P.S.G. and C.C.) and led as chief scientist the field experiment; provided the echo-sounder, current profiler, camera and winch used; analysed all the data presented except for the overturn detection analysis that was carried by P.S.G.; carried out the numerical simulations; made the initial discovery; initiated discussions with P.S.G. and C.C.; coordinated the work; produced the figures; made the literature review; wrote the text. P.S.G. co-organized and participated in the field experiment; provided the boat; piloted the boat; provided the temperature–salinity–depth profiler; carried out the overturn detection analysis; participated in several exchanges with D.B. and C.C. to discuss the analyses performed and the results obtained; critically reviewed and improved the text. C.C. co-organized and participated in the field experiment; participated in several exchanges with D.B. and P.S.G. to discuss the analyses performed and the results obtained; assisted D.B. with the literature review; critically reviewed and improved the text.

## Additional information

**Competing financial interests:** The authors declare no competing financial interests.

**Publisher's note**: 

