## [Peer Review file · Nature Communications]

Reviewers' comments:

Reviewer #1 (Remarks to the Author):

I enjoyed reading this paper, which describes a very nice set of observations of nonlinear waves (NLIW) and a frontally-forced intrusion in a fjord. The authors argue that the waves arise in a multi-step process involving i) a convergent front giving rise to ii) an intrusion at the depth of the thermocline that iii) radiates NLIW at its leading edge while also iv) generating K-H billows at its upper and lower edges. The authors use a) georectified camera from shore, b) acoustic backscatter, c) ADCP and d) CTD data from a small boat to measure the process, and run a 2D numerical model to isolate the physics. I find the observations convincing and the model very convincing that a process similar to i -> iv is occurring at this time and location in this particular fjord. They have therefore nicely demonstrated the existence of process i-iii, possibly for the first time, but iv has been observed in fjords before (Mickett et al 2004, Alford et al 2005, possibly also other work by Geyer et al?).

I do not find their arguments of the global importance of the process at all convincing. First, fjords are such a special geophysical situation (N about 50 times ocean thermocline values, strong tidal convergences, etc) that I am not sure of their relevance to the open ocean. Next, I found the description of KH billows and NLIW in the introduction a little out of touch with the general state of the art of mixing in the ocean, and logically a little odd. Specifically, they note that climate and regional models suffer from not knowing the mechanisms that trigger turbulence (true); however, their further assumption that what is needed is a new mechanism (lines 22-23) doesn't logically follow, to me at least. Third, if they are claiming that intrusions might be important for generating shear in the ocean (which has been claimed before), they should make some attempts at assessing their importance - and the vast literature on thermohaline intrusions in the ocean should probably be mentioned. Finally, I believe that there must be an error in their energy calculation, such that I was not convinced by the assertion that the process is energetically comparable to the South China Sea.

In summary, I find this a fairly clearly written documentation of very nice observations and a model of processes i-iv in a fjord, of which i-iii is new. I do not agree at all with their arguments for the global importance of the mechanisms. However, all is not lost for the publication at all: it is possible that with a completely rewritten introduction and these other comments addressed, it may well belong in Nature Communications, but I would recommend refocusing it as a nice documentation of a hitherto unobserved physical process that is likely important for coastal/fjord mixing rather than attempting to claim its global significance (ie read some of David Farmer's papers for examples of how to position such processes as important without making grandiose claims). If the editors do not agree that it belongs in Nature Communications, it would certainly make an excellent contribution to a specialized journal (GRL), subject to these same comments.

Specific additional comments (major marked with a star):

10: unclear. "their" is meant to refer to models, but it could refer to fine-scale processes.

18-19: This is a big stretch from the data presented.

22 and 26-27: In both of these cases, the assumption appears to be that our understanding is limited because of a limited number of mechanisms such that a new mechanism will help things.

32 uncovered: wrong word.

*43-48: This is not a very satisfying overturn analysis. The overturns mostly do not occur in the sheared region above the intrusion (or do they? Perhaps plot shear?) These details should be expanded and moved to the methods. Readers should be told how the data were processed and screened (Galbraith and Kelley?), what the minimum overturn you can detect is and that that is limiting the picture substantially. There is an apparent 4-m overturn in the echosounder near 250 m on the LHS of figure 2; it looks like a CTD went right through it. I do not think it is useful to compute a mean over 7 overturns spaced as widely as those reported to give one dissipation value. Finally, maybe just plot them with their actual extent? Or at least give a scale for the energy mentioned in the caption.

56-57: Frontal convergence: can you estimate if the convergence rates are consistent with the observed volume flux into the intrusion?

57: Characterized BY

58: entrainment

59: w is positive downwards? Also, what is the water depth? Not sure if I saw that anywhere.

60: Did you actually measure a density front with horizontal scales 10 m? Your CTD spacing was much more coarse than this. Perhaps you are inferring scales from the echosounder. In that case you should not say density jumps over 10 m, as you did not measure that.

60: What is the value of R_0 ?

61: What category do you mean?

65 How strong was the surface current?

68: A hypothesis

72: suggest -> suggests

77: The simulations appear to agree well with the observations and are very convincing.

90-92: I guess I don't really see that the main northbound waves are that different from river plume waves. Like the Nash/Moum case, the waves arise from a tidally-modulated convergent front. These waves do have an intermediate stage of a mid-depth intrusion. Is that so different? Please explain.

*95-108: I am unconvinced by this energy calculation and its comparison to the South China Sea. For starters, this is a model calculation. What is the energy in the observations (KE at least; PE might not be estimable with their measurements)? If we do take the model calculations at face value, it is a very strong wave (top 1% or so) compared to Shroyer et al's waves in coastal New Jersey (which should possibly be cited?). That's fine, but following Moum et al 2007 (which should definitely be cited), energy flux should be approximately cE or 7.3 kW/m , a factor of ten less than the 80 they report. Also, if the waves have never been observed at spring tide (perhaps stronger lateral shear, stronger mixing, or any number of other processes shut the process down) then it seems a little cavalier to simply scale with tidal forcing (another factor of 4).

By the way, I suppose that the ultimate energy source for these waves is the barotropic tide in the fjord? Would it be insightful to compare these numbers to see if they represent an appreciable sink of energy for it?

110: exercise

Review of
**“Generation of Kelvin-Helmholtz Billows and Solitons
by Frontally Forced Intrusions”**

by D. Bourgault, P. S. Galbraith and C. Chavanne

In this paper observations of internal solitary waves in Saquenay Fjord are presented. Using some numerical modeling for guidance the authors suggest that the observed solitary waves are generated by the adjustment of a submesoscale front which is phase locked to the semi-diurnal tide.

The observations are quite interesting however the modelling and associated interpretation are suspect. The boundary conditions are never stated however the statement that the simulation were “carried out for a flat-bottom rectangular domain of depth $H = 150$ m” suggests a rigid lid. If that is the case the initial conditions are incompatible with the boundary condition because a rigid lid together with the incompressibility approximation (their equation (2)) requires that the vertical integral of v be independent of y . This is not the case for the initial conditions used in the simulation which has a volume flux of $0.3 \times 14 - 0.4 \times 132 = -50.2 \text{ m}^2 \text{ s}^{-1}$ at the left boundary and $-0.8 \times 8 - 0.4 \times 142 = -63.2 \text{ m}^2 \text{ s}^{-1}$ at the right boundary. It is also qualitatively obvious: the vertical integral of v_y must be zero so any convergence in the upper layer must be balanced by divergence in the lower layer. Indeed, the horizontal velocity field at $t = 2$ minutes shown in Figure 4 appears to have such a divergence so the model has undergone some adjustment. At this time it appears, from the figure, that the flow in the upper layer to the left of the front has decreased in magnitude as has the flow in the lower layer to the right of the front. This adjustment makes the report values of the initial Richardson and Froude numbers pointless. It also raises questions about the reported energy and energy flux values (e.g., what was the background state?). The simulation should be redone with an appropriate initialization (including for w which appears to have been initialized with $w = 0$ here — there is no reason not to initialize with a dynamically consistent velocity field).

More information on how the energy and energy flux values should be provided. How was the background current taken into account? The formula used to calculate the energy flux given in line 191 is incorrect: if the vertical integral of v is non-zero, as is the case for the initial conditions on both sides of the front, then $\int v \rho g z dz$ depends on the coordinate system: that is you get different values if $z = 0$ at the surface or is $z = 0$ at the bottom. One should really be calculating the APE flux, e.g., equation (3) in the cited paper by Lamb and Nguyen.

I calculated some internal solitary waves using the initial fields to the right of the front (which as discussed above are probably not correct) and got an energy of

about 0.8 MJ/m for a wave of amplitude 13 m and a propagation speed of about 0.8 m/s. The energy in the wave is an order of magnitude smaller than the value reported here.

Overall there appears to be an interesting story to tell in explaining the observations but more work is required. The paper should be rejected but I encourage the authors to put together a convincing consistent story and resubmit.

Other suggestions:

1. I don't understand what is meant by "the shear source was uncovered" on line 32. Is the mechanism for the generated waves really that different from they way they are generated by a river plume? Here there is shear in the background field upstream of the front which may not occur for river plumes (but may?), but fundamentally wouldn't the generation process be the same?
2. The panels in the figures should be labeled so, for example, on line 57 you could refer to figure 2a.
3. It would be helpful to add some bars of length 1 km in the north-south and east-west directions to figure 1 so the reader has a sense of how far apart the observed waves are.
4. The expression for N on line 146 is missing a square root.
5. It is more conventional to have z and w positive upward so I suggest the authors do that although using z increasing downward is of course OK.
6. I think internal solitary wave is better terminology. Technically the waves are not solitons, which are solitary waves with a very special property.

A report on ‘Generation of Kelvin-Helmholtz Billows and Solitons by Frontally Forced Intrusions’ by Bourgault *et al.*

April 25, 2016

Abstract

The authors solve a sort of ‘Riemann problem’ for stratified flows. This problem is well known in gas dynamics and consists in finding an auto-similar solution of Euler equations of compressible fluids with discontinuous piece-wise constant initial data. Its non-stationary solution consists of shocks waves and simple waves separated by a contact discontinuity. Such a problem for stratified flows is much more complicated. Indeed, the discontinuous initial data here are given density and velocity profiles, approximately describing a forced frontal intrusion. Even if these profiles are ‘stable’ in the sense of the Miles theorem, the solution can be very complicated : it is not auto-similar, produces the solitary wave trains etc. In general, the paper is well written and can give an interesting information for a large community of scientists. I would suggest to accept it for publication at the condition that the remarks formulated below are taken into account.

Some remarks :

1. p. 9 line 146 : should be written N^2 instead of N .
2. Sometimes, for ‘prepared’ initial data, only one very regular wave can be obtained. This is the case of so-called ‘conjugate flows’ producing a bore or a table-top soliton in a stratified fluid (see B. Benjamin, J . Fluid Mech. (1966), Internal waves of finite amplitude and permanent form, vol. 25, part 2, pp. 241-270 and a recent paper by A. Kazakov, Conjugate shear flows of a weakly stratified fluid, Journal of Applied Mechanics and Technical Physics, Vol. 50, No. 2, pp. 235242, 2009). It is worth to cite these papers.

Response to Reviewer 1

Thank you for your encouraging and constructive comments that greatly contributed to improve the manuscript. We have made fairly major changes following your recommendations. There is perhaps only one point on which we may disagree but we hope you will be convinced by our response. It is about whether or not our results obtained in a fjord can be extrapolated to other geophysical situations. This is discussed in detail at point 2 below.

General comments

1. *They have therefore nicely demonstrated the existence of process i-iii, possibly for the first time, but iv has been observed in fjords before (Mickett et al 2004, Alford et al 2005, possibly other work by Geyer et al?)*

We agree. Thank you for the references. K-H billows driven by intermediate intrusions has indeed been reported before. Following this comment, we have decided to put much less emphasis on the generation of K-H billows and to stay much more focused on the generation of internal solitary waves. Accordingly, we have modified the title to: Generation of Internal Solitary Waves by Frontally Forced Intrusions in Geophysical Flows.

We have also added references to Geyer and Cannon (1982), Cannon et al. (1990), Mickett et al. (2004), Alford et al. (2005) and Belzile et al. (2016) for deep and intermediate intrusions found in fjords as well as a comment to illustrate how our pycnocline intrusion differs from those reported intrusions [lines: 105-107].

2. *I do not find their arguments of the global importance of the process at all convincing. First, fjords are such a special geophysical situation (N about 50 times ocean thermocline values, strong tidal convergences, etc). ... I do not agree at all with their arguments for the global importance of the mechanisms.*

This is the only point we must disagree and we hope Reviewer 1 will find our response convincing. It's not the difference in stratifications by itself, or that of the strength of tidal currents, that matters when comparing different environments. It's some nondimensional ratios such as the Rossby number (Ro), The Reynolds number (Re), the Richardson number (Ri), the Froude number (Fr), the ratio of wave amplitude to surface layer thickness (a/h_1), the ratio of the wavelength to surface layer thickness (λ/h_1), etc. None of those numbers we are reporting that characterize the conditions in the Saguenay Fjord are shockingly different than values that can be found in oceans, lakes or the atmosphere.

Another way to see this is that if only the stratification mattered then all lab experiments would be useless for understanding geophysical fluids. Take for example the low Reynolds number laboratory work of Sutherland et al. (2004) or Flynn et al. (2008). These studies are motivated by geophysical situations but addressed in the lab with stratification many orders of magnitude larger than any geophysical situation. Yet, extremely highly stratified, cm-scale laboratory experiments can acceptably be extrapolated to kilometer-scale natural environments as long as dynamic similarities are respected. This often works except for the Reynolds number and the width:depth aspect ratio that are generally difficult to respect in laboratory experiments. At least in the Saguenay Fjord, the Reynolds number and the width:depth aspect ratio are truly of geophysical relevance. With this in mind, our results can well be extrapolated to other dynamically and geometrically similar environments, at least as well as, or better than, laboratory experiments.

So, by itself, the argument that the results cannot be generalized because the stratification or currents in fjords are higher than other environments simply does not hold. We are convinced that Reviewer 1 will agree with us on this. So the comment really was that this was not convincingly expressed in our first version of the manuscript. Following this comment we have made more emphasis on nondimensional numbers and we are now more careful when we suggest that the results could be extrapolated by explicitly indicating that results should hold for *dynamically similar* environments. For example, this sentence can be found in the abstract: "... this result suggests that frontally forced intrusions may be important sources of internal solitary waves in regions of lakes, oceans and atmospheres that are dynamically similar to the situation examined here."

3. *Next, I found the description of KH billows and NLIW in the introduction a little out of touch with the general state of the art of mixing in the ocean, and logically a little odd. Specifically, they note that climate and regional models suffer from not knowing the mechanisms that trigger turbulence (true); however, their further assumption that what is needed is a new mechanism (lines 22-23) doesn't logically follow, to me at least.*

We agree with this comment. In response, we have completely rewritten the introduction and we have given the paper a different pitch by focusing much more on the fundamental problem of internal solitary waves generation by the gravitational collapse of mixed fluids. The motivations are now less pretentious than the previous version.

4. *Third, if they are claiming that intrusions might be important for generating shear in the ocean (which has been claimed before), they should make some attempts at assessing their importance - and the vast literature on thermohaline intrusions in the ocean should probably be mentioned.*

As mentioned above at point #1, the focus of the manuscript is not anymore so much on shear instability. Also, we've have decided not to attempt to extrapolate our results to intrusions in general and to stick to frontally forced intrusions. So this sentence: "...this result suggests that frontally forced intrusions, and perhaps intrusions more generally, may be important sources of internal solitary waves in geophysical fluids." has been replaced by "...this result suggests that frontally forced intrusions may be important sources of internal solitary waves in dynamically similar geophysical fluids...."

5. *Finally, I believe that there must be an error in their energy calculation...*

Yes, there was indeed an error. This was also pointed out by Reviewer 2. Please see our detailed response to Reviewer 2 below under section Energy flux.

Specific comments

1. *10: unclear. "their" is meant to refer to models, but it could refer to fine-scale processes.*
This sentence no longer exists.

2. *18-19: This is a big stretch from the data presented.*

We agree. See our response to points #2 and #3 above.

3. *22 and 26-27: In both of these cases, the assumption appears to be that our understanding is limited because of a limited number of mechanisms such that a new mechanism will help things.*

The introduction has been completely rewritten. This should not be an issue anymore.

4. *32 uncovered: wrong word.*

This sentence no longer exists.

5. **43-48a: This is not a very satisfying overturn analysis. The overturns mostly do not occur in the sheared region above the intrusion (or do they? Perhaps plot shear?).*

A new figure showing the shear and the reduced stratification has been added (Fig. 4).

6. **43-48b: These details should be expanded and moved to the methods. Readers should be told how the data were processed and screened (Galbraith and Kelley?), what the minimum overturn you can detect is and that that is limiting the picture substantially.*

These informations are now included in the Methods [lines 239-243].

7. **43-48c: There is an apparent 4-m overturn in the echosounder near 250 m on the LHS of figure 2; it looks like a CTD went right through it.*

No, this overturn was not detected by the CTD. It didn't go right through it. All overturns detected are plotted as 'Zs' on Figure 2.

8. **43-48d: I do not think it is useful to compute a mean over 7 overturns spaced as widely as those reported to give one dissipation value.*

We agree, but at the same time this is all we have. To address this comment we now simply give an order of magnitude rather than an actual value ($\mathcal{O}(\bar{\epsilon}) \sim 10^{-5} \text{ W kg}^{-1}$). This is not central to the demonstration.

9. **43-48e: Finally, maybe just plot them with their actual extent? Or at least give a scale for the energy mentioned in the caption.*

The size of the Z on Figure 2 are proportional to the log of the available potential energy P of the fluctuations. So there is already a scale for the energy in the figure. The largest Z has $P = 3 \times 10^{-4} \text{ W kg}^{-1}$ and the smallest $P = 4 \times 10^{-6}$. This information was already in the figure caption so no modifications made.

10. *56-57: Frontal convergence: can you estimate if the convergence rates are consistent with the observed volume flux into the intrusion?*

This is automatically done by the numerical model. The numerical model is initialized with a condition close to the observed convergent front and this produces an intrusion with a comparable velocity and thickness as the observations. So yes, the fact that the numerical results convincingly reproduces the main characteristics of the intrusion indicates that the observed convergence rate are consistent with the volume flux into the intrusion. A brief remark has been added to the text to emphasize this aspect of the simulation [lines: 156-158].

11. *57: Characterized BY*

Fixed.

12. *58: entrainment*

Fixed.

13. *59: w is positive downwards?*

Yes, it was. This is how the model is coded (Bourgault and Kelley, 2004). Reviewer #2 also made that remark so we now report all equations and figures with z being positive upward.

14. *59: Also, what is the water depth? Not sure if I saw that anywhere.*

This is now stated: "The total depth along this transect varies between $100 < H < 150 \text{ m}$ " [line: 65].

15. *60: Did you actually measure a density front with horizontal scales 10 m? Your CTD spacing was much more coarse than this. Perhaps you are inferring scales from the echosounder. In that case you should not say density jumps over 10 m, as you did not measure that.*

No, we did not measure the actual density jump at that scale. Yes, the scale was inferred from the echo-sounder and ADCP and we actually re-evaluated it to be 20 m. The sentence has been modified accordingly [lines: 109-113].

16. *60: What is the value of Ro ?*

Now explicitly stated as $Ro = v/(fL) = 2 \times 10^2 \gg 1$.

17. *61: What category do you mean?*

We meant the category of submesoscale fronts. Perhaps the word “category” was not appropriate in this context. We changed it to “type”.

18. *90-92: I guess I don't really see that the main northbound waves are that different from river plume waves.... Please explain.*

The new introduction should now make this a lot clearer.

19. **95-108a: I am unconvinced by this energy calculation and its comparison to the South China Sea. For starters, this is a model calculation. What is the energy in the observations (KE at least; PE might not be estimable with their measurements)? If we do take the model calculations at face value, it is a very strong wave (top 1% or so) compared to Shroyer et al's waves in coastal New Jersey (which should possibly be cited?).*

The energy calculation has been completely revised. See below our detailed response to Reviewer 2 about this. We have also removed the comparison with the South China Sea which was indeed irrelevant. We now give an estimation of the wave energy seen in Figure 2a with the help of the model rather than using the methods of Moum et al. (2007). This is just another acceptable way of estimating wave energy. Basically our 2D non-hydrostatic and nonlinear model is set with conditions that are close to the background sheared environment and produces wave of various amplitudes. We calculated from the model the energy of a wave that had the same amplitude as the wave see in Figure 2a. This gives an total mechanical energy (kinetic + potential) of $E = 0.8 \text{ MJ m}^{-1}$.

We have decided not to cite Moum et al. (2007) because we feel that the subject of that paper is not directly related to our study. Moum et al. (2007) is about shoaling waves of elevation. If we'd cite Moum et al. (2007), then we would have to cite many other references on that subject. Moum et al. (2007) is indirectly cited through the reference made to the review of Lamb (2014).

20. **95-108b: That's fine, but following Moum et al 2007 (which should definitely be cited), energy flux should be approximately cE or 7.3 kW/m , a factor of ten less than the 80 they report.*

There may be confusion here. The 80 kW m^{-1} we reported in the previous version was the peak maximum depth-integrated flux value. The revised energy calculation now gives a peak value for the leading wave of 37 kW m^{-1} (see the new Figure 7 that shows explicitly the energy flux).

Moum et al. (2007) used $c < E >$ where $< >$ is an integral over depth and time such that the units of $c < E >$ should be in J m^{-1} not W m^{-1} . But Reviewer 1 refers to cE in W m^{-1} (well, kW m^{-1}) which is confusing too because following Moum et al. (2007) cE should be in W m^{-2} (i.e. $\text{m s}^{-1} \times \text{J m}^{-3}$). E in Moum et al. (2007) is an energy density (per unit volume). So there's confusion here.

To give a sense of scale we could compare the total wave energy calculated from integrating in time the energy flux over the leading wave simulated (see Figure 7) with the energy of a KdV wave, that is

$$E_{KdV} = \frac{4}{3}g\Delta\rho a^2(\lambda/2). \quad (1)$$

The manuscript reports that this first wave has total energy $E_1^+ = 2.3 \text{ MJ m}^{-1} \sim 1 \text{ MJ m}^{-1}$ [line: 167]. That wave is also reported to have the following characteristics: $(\Delta\rho, a, \lambda) = (16 \text{ kg m}^{-3}, 14.1 \text{ m}, 70 \text{ m})$ [lines: 166-167]. This gives $E_{KdV}^+ = 1.5 \text{ MJ m}^{-1} \sim 1 \text{ MJ m}^{-1}$. This back-of-the-envelope calculation suggests that, this time, the value reported makes sense.

21. *Also, if the waves have never been observed at spring tide (perhaps stronger lateral shear, stronger mixing, or any number of other processes shut the process down) then it seems a little cavalier to simply scale with tidal forcing (another factor of 4).*

Internal solitary waves had been seen in this environment during spring tides. One such wavetrain is reported by Bourgault et al. (2011) which was indeed captured 3 days after a spring tide maximum (observations on the 5 July 2007, max spring tide on the 2). We cannot tell whether that wavetrain had been generated by the mechanism we report here. But we agree that we don't have much to support this argument. We have removed this argument.

22. *By the way, I suppose that the ultimate energy source for these waves is the barotropic tide in the fjord? Would it be insightful to compare these numbers to see if they represent an appreciable sink of energy for it?*

Yes, it would be insightful. However, at this point, that would also be, to take the Reviewer's expression, a little cavalier to attempt to extrapolate such an energy budget with a single evidence. We feel we have too little information to do this. Furthermore, the wave energy would need to be integrated horizontally, along wave crest to get total wave energy and this information is not always trivial to get, unless an assumption is made that the waves always occupy the entire width of the fjord but we already know that this would be difficult to justify. So thanks for the suggestion but we've decided not to attempt this sort of extrapolation for now.

23. *110: exercise*

Corrected.

24. *150-152: I couldn't parse this.*

Please see Baker and Gibson (1987) for details. No changes made.

Response to Reviewer 2

Thank you for your critical review and comments, all relevant, constructive and encouraging. We agree with your comments and they have all led to significant improvements to the manuscript as detailed below.

Initial condition

The model has a free-surface. This is now stated in the Methods.

Indeed, the model underwent some initial adjustment for the reasons explained by the reviewer. This was stated on line 185 of the original version: “This long coarse resolution domain acts as a sponge layer that numerically absorbs *barotropic waves that escape during the initial adjustment ...*”. So, we were aware of this and it didn’t seem to be much of a problem as the model could deal with this. However, we agree that this was not very elegant. Also, and perhaps more importantly, the reviewer has made us realize that this made “the report values of the initial Richardson and Froude numbers pointless.” We agree with this and we thank the reviewer for having pointed that out.

We have modified the initial condition for the horizontal velocity accordingly. The initial velocity field on each side of the front is now set such that

$$\int_{-H}^0 v_- dz = \int_{-H}^0 v_+ dz. \quad (2)$$

The new initial conditions are now:

$$v_- = v_{01} + \frac{v_1 - v_{01}}{2} \left[1 + \tanh \left(\frac{z + h_1}{d_1} \right) \right], \quad (3)$$

$$v_+ = v_{02} + \frac{v_2 - v_{02}}{2} \left[1 + \tanh \left(\frac{z + h_2}{d_2} \right) \right], \quad (4)$$

with $(v_{01}, v_{02}, v_1, v_2) = (-0.4000, -0.3088, 0.3000, -0.8000) \text{ m s}^{-1}$. The only difference from the original version is that the flow in the bottom layer is now set with two parameters (v_{01}, v_{02}) rather than only one before (v_0) .

The procedure for determining those parameters was first to set the values for v_{01}, v_1, v_2 that match well the field observations (Fig. 5 in the manuscript) and then solve equation 2 for v_{02} .

The resulting analytical expression is quite complex and is omitted in the manuscript. It is only presented here for the benefit of Reviewer 2,

$$\begin{aligned}
v_{02} = & \left[\frac{H}{2}(v_1 + v_{01} - v_2) \right. \\
& + \frac{d_1(v_{01} - v_1)}{2} (\log[\cosh((H - h_1)/d_1)] - \log[\cosh(-h_1/d_1)]) \\
& + \left. \frac{d_2 v_2}{2} (\log[\cosh((H - h_2)/d_2)] - \log[\cosh(-h_2/d_2)]) \right] / \\
& \left[\frac{H}{2} + \frac{d_2}{2} (\log[\cosh((H - h_2)/d_2)] - \log[\cosh(-h_2/d_2)]) \right]. \tag{5}
\end{aligned}$$

The initial vertical velocity is prescribed by integrating the continuity equation, that is,

$$w(y, z, t = 0) = - \int_{-H}^z \frac{\partial v}{\partial y} dz. \tag{6}$$

Figure 2 in the manuscript only shows the first 50 m of the water column. For the convenience of Reviewer 2, the figure below shows the initial vertical velocity over the entire depth.

Energy flux

The reviewer is right. Our energy flux calculation was incorrect. It is not so much related to the previous point. It's because our wave energy flux calculation was including a large portion of the kinetic energy of the background state. We (well, Bourgault) thought we (he) had filtered that part arising from the background field with an “envelope filter” (see line 192 of the original manuscript) but this was incorrect. We have reconsidered the energy flux calculation on each side of the front at $y = -1$ km (‘-’ subscript) and $y = 1$ km (‘+’ subscript) and it is now calculated as

$$F_- = \int_{-H}^{\eta} (v - v_-) \left[p_d + \frac{\rho_0}{2} [(v - v_-)^2 + w^2] + E_a \right] dz, \tag{7}$$

and

$$F_+ = \int_{-H}^{\eta} (v - v_+) \left[p_d + \frac{\rho_0}{2} [(v - v_+)^2 + w^2] + E_a \right] dz \tag{8}$$

where p_d is the sum of the hydrostatic and nonhydrostatic pressure and E_a is the available potential energy as defined in Lamb and Nguyen (2009) (their equation 16). The results now make more sense. With this calculation the leading wave of amplitude $a = 14.1$ m (see Figure 7 in the manuscript, the wave flux around $t = 20$ min) has energy $E_1^+ = 2.3 \text{ MJ m}^{-1}$ [line:

167].

In the first version the energy flux was calculated as

$$F = \int_{-H}^{\eta} v \left[p_d + \frac{\rho_0}{2} [v^2 + w^2] + E_a \right] dz, \quad (9)$$

and then incorrectly filtered.

Other suggestions

1. (a) *I don't understand what is meant by "the shear source was uncovered" on line 32.*
That sentence has been removed. By that we meant that we had identified the origin of the shear source. Most often shear layers and Kelvin-Helmholtz billows are observed but the cause of the shear is not always trivial to determine. Here, we thought we had found the origin of the shear.
- (b) *Is the mechanism for the generated waves really that different from they way they are generated by a river plume? Here there is shear in the background field upstream of the front which may not occur for river plumes (but may?), but fundamentally wouldn't the generation process be the same?*
The new introduction should now make this a lot clearer.
2. *The panels in the figures should be labeled so, for example, on line 57 you could refer to figure 2a.*
Good idea. Done.
3. *It would be helpful to add some bars of length 1 km in the north-south and east-west directions to figure 1 so the reader has a sense of how far apart the observed waves are.*
Good idea. Done in the east-west direction. No need for another scale in the north-south direction with this Mercator projection over such a small region.
4. *The expression for N on line 146 is missing a square root.*
Corrected. Thank you for pointing that out. It was only a typographical error. The calculation was done correctly.
5. *It is more conventional to have z and w positive upward so I suggest the authors do that although using z increasing downward is of course OK.*
Ok. Done. Equations and figures are now consistent with z being positive upward.

6. *I think internal solitary wave is better terminology. Technically the waves are not solitons, which are solitary waves with a very special property.*

We agree. We had used the term soliton simply to lighten the text in order to avoid a long name such as internal solitary waves or the use of an unpleasant acronym such as ISW. Solitons had been used widely in the oceanic literature and even recently (Apel et al., 2007) but we agree that the term is now not so much used. We've changed the text by *internal solitary waves* or sometime simply by *waves*.

Figure 1: Initial condition for w , positive upward.

Response to Reviewer 3

Thank you for your positive comments.

Some remarks

1. Corrected. Thank you.
2. References to Benjamin (1966) and to Kazakov (2009) have been added [line: 16].

References

- Alford, M.H., Gregg, M.C., D'Asaro, E.A., 2005. Mixing, 3D mapping, and lagrangian evolution of a thermohaline intrusion. *J. Phys. Oceanogr.* 35, 1689–1711.
- Apel, J.R., Ostrovsky, L.A., Stepanyants, Y.A., Lynch, J.F., 2007. Internal solitons in the ocean and their effect on underwater sound. *J. Acoust. Soc. Am.* 121, 695–722.
- Baker, M.A., Gibson, C.H., 1987. Sampling turbulence in the stratified ocean: statistical consequences of strong intermittency. *J. Phys. Oceanogr.* 17, 1817–1836.
- Belzile, M., Galbraith, P.S., Bourgault, D., 2016. Water renewals in the Saguenay Fjord. *J. Geophys. Res. Oceans* 121, 638–657.
- Benjamin, T.B., 1966. Internal waves of finite amplitude and permanent form. *J. Fluid Mech.* 25, 241–270.
- Bourgault, D., Janes, D.C., Galbraith, P.S., 2011. Observations of a large-amplitude internal wavetrain and its reflection off a steep slope. *J. Phys. Oceanogr.* 41, 586–600. doi:10.1175/2010JP04464.1.
- Bourgault, D., Kelley, D.E., 2004. A laterally averaged nonhydrostatic ocean model. *J. Atmos. Oceanic Technol.* 21, 1910–1924.
- Cannon, G.A., Holbrook, J.R., Paskinski, D.J., 1990. Variations in the onset of bottom water intrusions over the entrance sill of a fjord. *Estuaries* 13, 31–42.
- Flynn, M., Boubarne, T., Linden, P., 2008. The dynamics of steady, partial-depth intrusive gravity currents. *Atmos.-Ocean* 46, 421–432.
- Geyer, W.R., Cannon, G.A., 1982. Sill processes related to deep water renewal in a fjord. *J. Geophys. Res.* 87, 7985–7996.
- Kazakov, A.Y., 2009. Conjugate shear flows of a weakly stratified fluid. *J. Appl. Mech. Tech. Phys.* 50, 235–242.
- Lamb, K.G., 2014. Internal wave breaking and dissipation mechanisms on the continental slope/shelf. *Annu. Rev. Fluid Mech.* 46, 231–254.
- Lamb, K.G., Nguyen, V.T., 2009. Calculating energy flux in internal solitary waves with an application to reflectance. *J. Phys. Oceanogr.* 39, 559–580.

- Mickett, J., Gregg, M., Seim, H., 2004. Direct measurements of diapycnal mixing in a fjord reach - Puget Sound's Main Basin. *Estuar. Coast. Shelf Sci.* 59, 539–558.
- Moum, J.N., Klymak, J.M., Nash, J., Perlin, A., Smyth, W.D., 2007. Energy transport by nonlinear internal waves. *J. Phys. Oceanogr.* 37, 1968–1988.
- Sutherland, B.R., Kyba, P.J., Flynn, M.R., 2004. Intrusive gravity currents in two-layer fluids. *J. Fluid Mech.* 514, 327–353.

Reviewer #1 (Remarks to the Author):

As in my first review, I enjoyed reading this paper describing a nicely observed and interpreted geophysical phenomenon. I appreciate the authors' responses to some of my concerns. On reflection, I believe that this paper could be an important demonstration of intrusion-forced nonlinear internal waves. These are possibly similar to the frontally-forced bores reported at the Ocean Sciences meeting in New Orleans by Moum and colleagues. Since the parameter space is vast, as the authors point out, the paper should be published and is suitable for Nature Communications. I do have a few remaining comments, but I do not need to see the manuscript again. Best of luck to you.

I didn't really need the authors to educate me on nondimensional parameters in their response to my comment in my first review. My comment was intended to mean, of course, not that the greater stratification intrinsically matters but that the dynamical regime in fjords is different than nearly anywhere in the open ocean. Reexpressing my original objection in nondimensional numbers, the authors' computed Rossby number of 200 is at least a factor of ten anything observed in open-ocean submesoscale fields.

I therefore continue to believe that the authors still need to be careful in their statements of the global significance of the process. Better might be to document the phenomena observed, state the possible links to NLIW observed in the ocean (Nash/Moum, Moum et al 2016 reports from OS16 if they have anything written up yet), and let readers decide the importance.

I would also caution the authors about their new conclusion that things are in non equilibrium. While the model runs are suggestive, I think that concluding that anything is in equilibrium or non-equilibrium from these data is a big stretch.

My remaining comments are minor. I do note a large number of typos in the manuscript, including 5 in the abstract (lines 2, 4, 4, 11, 12).

line 16: These are a rather odd set of references for observations of ISW.

20: Reference 4 is not an observational paper as cited.

21: As I said in my first review, understanding of their origin is not elusive because of too few mechanisms. I find this a really odd statement.

23: I'd attempt to broaden this list of observations of topographically-generated ISW somewhat.

243-245: The dissipation is estimated from 11 overturns which are the extreme high-dissipation tail of the distribution. Therefore, it's not appropriate to assume log normality and employ Baker and Gibson. Much more relevant would be Gregg (1993). In any case, I agree with their wise choice to simply give an order of magnitude estimate given the extreme sparsity of the overturn data, which would seem to me to avoid the need for this correction at all.

332: Capitals needed.

Review of
**“Generation of Internal Solitary Waves by Frontally Forced Intrusions in
 Geophysical Flows”**

by D. Bourgault, P. S. Galbraith and C. Chavanne

The revised paper is greatly improved. I do have some concerns regarding the energy calculation as described in the appendix. Other than the paper is very nice. It is interesting and topical.

1. First to calculate E_a a reference density is needed. What reference density is used? Is the background density far to the right/left used to calculate the APE of the rightward/leftward propagating waves or is the density field in the whole domain sorted to determine the reference density? I think the former could be justified. The reference density used to compute the APE should be specified.
2. The pressure p_d in equations (15) and (16) is a perturbation pressure: it does not include the undisturbed hydrostatic pressure (though it does include a hydrostatic contribution). Should mention this.
3. A more significant concern is the calculation of the energy flux. A conservation law has the form

$$\frac{\partial Q}{\partial t} + \vec{\nabla} \cdot \vec{F} = 0 \quad (1)$$

where Q is the conserved quantity and \vec{F} the flux. For a conservation law the change of Q in a domain is easily written in terms of fluxes through the boundaries. For the incompressible Euler equations under the Boussinesq approximation we have the following energy equation in conservation form:

$$\frac{\partial}{\partial t} (E_k + E_a) + \vec{\nabla} \cdot (\vec{u}(p_d + E_k + E_a)) = 0 \quad (2)$$

where, for the 2D flow in the y - z plane as in this manuscript, $\vec{u} = (v, w)$ is the total velocity vector (background plus perturbation) and

$$E_k = \frac{\rho_0}{2} \vec{u} \cdot \vec{u} \quad (3)$$

is the total kinetic energy.

Separating v into a background field $\bar{V}(z)$ and a perturbation $v'(y, z, t)$, the y -component of the kinetic energy flux term is then

$$vE_k = \frac{\rho_0}{2} (\bar{V}^3 + 3\bar{V}^2v' + 3\bar{V}v'^2 + v'^3 + v'w^2). \quad (4)$$

The first part can be ignored as it is independent of y leaving

$$vE_k = \frac{\rho_0}{2} \left(3\bar{V}^2 v' + 3\bar{V} v'^2 + v'^3 + v' w^2 \right). \quad (5)$$

The authors ignore the first two terms and use

$$vE_k = v' \frac{\rho_0}{2} \left(v'^2 + w^2 \right) \quad (6)$$

in their equations (15) and (16). The difficulty is that there is no conservation law that has this kinetic energy flux. Defining the perturbation kinetic energy, which seems to be what the authors have in mind, as

$$E'_k = \frac{\rho_0}{2} \left(v'^2 + w^2 \right) \quad (7)$$

leads to the following equation:

$$\frac{\partial}{\partial t} \left(E'_k + E_a \right) + \vec{\nabla} \cdot \left(\vec{u} (p_d + E'_k + E_a) \right) + \frac{d\bar{V}}{dz} v' w = 0 \quad (8)$$

which is not in conservation form. The term $\frac{d\bar{V}}{dz} v' w$ is a familiar one in turbulence theory (after suitably averaging): it transfers energy from the perturbation field and the mean flow. Here it is responsible for transferring energy between the wave field and the mean flow. The perturbation wave energy defined using E'_k is not conserved.

The authors justify use of their method to calculate the kinetic energy flux by citing Martin *et al.* (2006) who use the same method (as do others). These authors simply state that that is how they are calculating wave energy (not fluxes as in the current manuscript) but they do not justify their choice so this is not a great reference to use.

Defining the perturbation kinetic energy using E'_k is justifiable when studying internal wave packets propagating through a slowly varying medium. In this case the wave kinetic energy is computed in a reference frame moving with the local background flow and wave packets can gain or lose energy via an exchange with the background flow. In particular, wave energy is not conserved. Wave action, a conserved quantity, is used instead. Internal solitary waves are a different beast. The wave has a structure that spans the full water column. It is in no sense a wave propagating vertically through a slowly varying background flow so I don't think use of E'_k can be justified in this context.

The authors estimate wave energy by integrating their energy flux as a wave crosses a line $y = \text{constant}$ (see their line 171). For the conservation law (2)

the wave energy can be correctly calculated in this manner. For the perturbation energy in (8) this can not be done without properly taking into the energy transfer term.

Minor Comments/suggesting rewording

1. Line 2: “intruding *into* two-layer ...”
2. Lines 11–12. Suggest “... when forced, do not need to be in non-equilibrium to generate large amplitude ...”. Strictly speaking the equilibrium state is an equilibrium state for a fluid at rest is it not? So perhaps saying it does not have to be in a non-equilibrium state to generate waves is maybe a bit misleading.
3. Line 18: “... waves are widespread and thought to be energetic” (change as they are known to be widespread, not simply thought to be).
4. Line 23. What do you mean by ‘subharmonic interactions of vortices’? The cited paper doesn’t mention vortices. It mentions subharmonic interaction and instabilities. I think the example in this paper is more complicated because flow over the sill is continually supplying energy to the wave field while varying in time so not just a simple energy transfer from one frequency to another.
5. Lines 28–30. The paper cited regarding ISW generation by ‘river plumes’, Nash and Moum (2005), is for a single river. Should either say ‘generated by the Columbia River plume’ or provide evidence for generation in other river plumes (I know it occurs because I have seen it for a small creek but I don’t know of any references off hand).
6. Line 53: “We report here *on* geophysical by non-equilibrium ... gravity currents ...” (delete the ‘a’ in front of non-equilibrium).
7. Line 65: “The total depth H ... between 100 and 150 m”.
8. Line 133: “This also explains why ...”
9. Line 179: ‘energetic’ (and on line 185 too).
10. Line 187: “... do not have to have a non-equilibrium stratification to generate large ...”

11. Caption for figure 1, 3rd line: “ ... as being evidence ...” or “as providing further evidence ...”.
12. Caption for figure 2: ‘Helmholtz’
13. Line 210: “from *an* 8-m research ...”
14. Line 308: “These energy ...”

Response to Reviewer 1

1. *I therefore continue to believe that the authors still need to be careful in their statements of the global significance of the process. Better might be to document the phenomena observed, state the possible links to NLIW observed in the ocean (Nash/Moum, Moum et al 2016 reports from OS16 if they have anything written up yet), and let readers decide the importance.*

Following this recommendation, we have changed this sentence of the abstract:

Given the widespread occurrences and importance of geophysical converging fronts, this result suggests that frontally forced intrusions may be important sources of internal solitary waves in regions of lakes, oceans and atmospheres that are dynamically similar to the situation examined here.

by this one:

These observational and modelling results suggest that frontally forced intrusions may represent additional and unexpected sources of internal solitary waves in regions of lakes, oceans and atmospheres that are dynamically similar to the situation examined here in the Saguenay Fjord. [lines: 13-15]

A similar sentence that appeared at the end of the second to last paragraph of the manuscript has simply been removed.

2. *I would also caution the authors about their new conclusion that things are in non equilibrium. While the model runs are suggestive, I think that concluding that anything is in equilibrium or non-equilibrium from these data is a big stretch.*

Following this comment, we have changed this sentence:

We report here geophysical observations of the generation of internal solitary waves by a non-equilibrium, partial-depth and frontally-forced interfacial gravity currents intruding into a quasi two-layer and vertically sheared ambient.

to:

... generation of internal solitary waves by what can be interpreted as being a non-equilibrium, partial-depth ... [lines: 52-53]

3. *I do note a large number of typos in the manuscript, including 5 in the abstract (lines 2, 4, 4, 11, 12).*

The quality of the english should now be improved.

4. *line 16: These are a rather odd set of references for observations of ISW.*

We have removed the reference to Kazakov (2009) which indeed felt out of place. We have however kept the other references (Benjamin, 1966; Miles, 1980; Helfrich and Melville, 2006). Those references are not so much listed to refer to observations of ISW but rather to refer to the fundamentals of ISW as a general introduction to such particular waves.

5. *20: Reference 4 is not an observational paper as cited.*

Reference 4 was a reference to Helfrich and Melville (2006). We agree that this reference is not specifically about observations of internal waves. It is nevertheless a review paper that does include many references to field observations of internal solitary waves. We have kept that reference but we also added a reference to Jackson and Apel (2004). [line: 20]

6. *21: As I said in my first review, understanding of their origin is not elusive because of too few mechanisms. I find this a really odd statement.*

Following this comment, we have changed this sentence:

Although these waves, once formed, are routinely observed in lakes, oceans and the atmosphere, their origin often remains elusive as there is only a limited number of known generation mechanisms to guide the interpretation of complex and sparse field measurements.

to:

Although these waves, once formed, are routinely observed in lakes, oceans and the atmosphere, their origin often remains elusive. [lines: 20-21]

7. *23: I'd attempt to broaden this list of observations of topographically-generated ISW somewhat.*

There were already 6 major references to topographically-generated ISW (Maxworthy, 1979; Armi and Farmer, 1988; Farmer and Armi, 1988; Cummins et al., 2003; Stastna and Peltier, 2005; Jackson et al., 2012) including a reference to Jackson et al. (2012); a review paper that itself include several more references on the subject. We have added to the list a 7th reference to Apel et al. (1985). [line: 22]

8. *332: Capitals needed.*

Corrected.

Response to Reviewer 2

1. *First to calculate E_a a reference density is needed. What reference density is used? ...*
The background density far to the right/left is used to calculate E_a . This is now stated.
[lines: 301-303]
2. *The pressure p_d in equations (15) and (16) is a perturbation pressure: it does not include the undisturbed hydrostatic pressure (though it does include a hydrostatic contribution). Should mention this.*
This is now mentioned. [line: 301]
3. *A more significant concern is the calculation of the energy flux....*
We now follow the Reviewer's recommendation and calculate the perturbation energy flux as recommended, with a reference to Lamb (2010). [line: 298-306]

Minor comments

1. *Line 2: "intruding into two-layer ..."*
Done.
2. *Lines 11-12. Suggest "... when forced, do not need to be in non-equilibrium to generate large amplitude ...". Strictly speaking the equilibrium state is an equilibrium state for a fluid at rest is it not? So perhaps saying it does not have to be in a non-equilibrium state to generate waves is maybe a bit misleading.*
This sentence:

...shows that intruding gravity currents, when forced, do not require to be in non-equilibrium for generating large amplitude internal solitary waves.

has been replaced with:

... shows that forced intruding gravity currents can generate large amplitude internal solitary waves even when the initial state of rest is in equilibrium.

[lines: 11-13]
3. *Line 18: "... waves are widespread and thought to be energetic" (change as they are known to be widespread, not simply thought to be).*
Modified as suggested. [line: 18]

4. *Line 23. What do you mean by “subharmonic interactions of vortices”?...*
Yes, it should only be “subharmonic interactions”. Corrected. [line: 22]
5. *Lines 28-30. The paper cited regarding ISW generation by “river plumes”, Nash and Moum (2005), is for a single river. Should either say “generated by the Columbia River plume...”*
Modified as suggested. [lines: 27-29]
6. *Line 65: “The total depth H ... between 100 and 150 m”.*
Corrected.
7. *Line 53: “We report here on geophysical by non-equilibrium ... gravity currents ...” (delete the “a” in front of non-equilibrium).*
Corrected.
8. *Line 133: “This also explains why ...”*
Corrected.
9. *Line 179: “energetic” (and on line 185 too).*
Corrected.
10. *Line 187: “... do not have to have a non-equilibrium stratification to generate large ...”*
The sentence has been rephrased to:

These results show that intruding gravity currents, when forced, can generate large amplitude internal solitary waves even when the initial state at rest is in equilibrium. [lines: 187-189]
11. *Caption for figure 1, 3rd line: “ ... as being evidence ...” or “as providing further evidence ...”.*
Corrected.
12. *Caption for figure 2: “Helmholtz”*
Corrected.
13. *Line 210: “from an 8-m research ...”*
Corrected.
14. *Line 308: “These energy ...’*
Corrected.

References

- Apel, J.R., Holbrook, J.R., Liu, A.K., Tsai, J.J., 1985. The Sulu Sea internal soliton experiment. *J. Phys. Oceanogr.* 15, 1625–1651.
- Armi, L., Farmer, D.M., 1988. The flow of Mediterranean water through the Strait of Gibraltar. *Progr. Oceanogr.* 21, 1–105.
- Benjamin, T.B., 1966. Internal waves of finite amplitude and permanent form. *J. Fluid Mech.* 25, 241–270.
- Cummins, P.F., Vagle, S., Armi, L., Farmer, D.M., 2003. Stratified flow over topography: upstream influence and generation of nonlinear internal waves. *Proc. R. Soc. Lond. A* 459, 1467–1487.
- Farmer, D.M., Armi, L., 1988. The flow of Atlantic water through the Strait of Gibraltar. *Progr. Oceanogr.* 21, 1–105.
- Helfrich, K.R., Melville, W.K., 2006. Long nonlinear internal waves. *Annu. Rev. Fluid Mech.* 38, 395–425.
- Jackson, C.R., Apel, J., 2004. An atlas of internal solitary-like waves and their properties. Second Edition. Technical Report Code 322PO, Contract N00014-03-C-0176. Office of Naval Research.
- Jackson, C.R., Da Silva, J.C., Jeans, G., 2012. The generation of nonlinear internal waves. *Oceanography* 25, 108–123.
- Kazakov, A.Y., 2009. Conjugate shear flows of a weakly stratified fluid. *J. Appl. Mech. Tech. Phys.* 50, 235–242.
- Lamb, K., 2010. Energetics of internal solitary waves in a background sheared current. *Non-linear Proc. Geoph.* 17, 553–568.
- Maxworthy, T., 1979. A note on the internal solitary waves produced by tidal flow over a three-dimensional ridge. *J. Geophys. Res.* 84, 338–346.
- Miles, J.W., 1980. Solitary waves. *Annu. Rev. Fluid Mech.* 12, 11–43.
- Stastna, M., Peltier, W.R., 2005. On the resonant generation of large-amplitude internal solitary and solitary-like waves. *J. Fluid Mech.* 543, 267–292.

REVIEWERS' COMMENTS:

Reviewer #2 (Remarks to the Author):

I am happy with the revised manuscript and recommend it be accepted. I just a few minor comments:

Line 26: 'wave generation' or 'waves generated'

Lines 47-48: What exactly is unsolved? This sentence is too vague and maybe too strong.

Line 53: define partial-depth

Line 197: The definition of equilibrium involves the stratification only. It does not take into account the background currents so I suggest using the term 'non-equilibrium stratifications' here and throughout. As I mentioned earlier I don't agree with using the definition of non-equilibrium for a fluid at rest and then using the same definition when there is a background current. That changes the energetics of the flow.

Response to Reviewer 2

1. *Line 26: 'wave generation' or 'waves generated'.*

Changed to 'waves generated'. [line: 29]

2. *Lines 47-48: What exactly is unsolved? This sentence is too vague and maybe too strong.*

This sentence:

The generation of internal solitary waves by non-equilibrium interfacial gravity currents is still an unsolved problem whose parameter space has not yet been fully explored.

has been replaced with:

The generation of internal solitary waves by non-equilibrium interfacial gravity currents is not yet fully understood partly because the wide parameter space that defines this problem has only been partially explored. [lines: 48-49]

3. *Line 53: define partial-depth.*

Partial-depth is now defined in the paragraph just before with the addition of the following sentence:

For example, the case where the lock water initially occupies only a fraction of the total water depth, a situation referred to as partial-depth intrusion, remains little studied, although likely of greater environmental significance. [lines: 50-52]

4. *Line 197: The definition of equilibrium involves the stratification only. It does not take into account the background currents so I suggest using the term 'non-equilibrium stratifications' here and throughout. As I mentioned earlier I don't agree with using the definition of non-equilibrium for a fluid at rest and then using the same definition when there is a background current. That changes the energetics of the flow.*

Every appearances of the term 'non-equilibrium' has been modified accordingly. Either we use 'non-equilibrium stratification' as suggested or some other phrases to clarify when there might be ambiguity. For example, on lines 153-154 we wrote "For partial-depth and initially motionless intrusions, the degree of non-equilibrium can be characterized by..." So we added 'and initially motionless' for clarity.